# Knowledge, Attitude, Practice, and Perceived Barriers for the Compliance of Standard Precautions among Medical and Nursing Students in Central India

**DOI:** 10.3390/ijerph20085487

**Published:** 2023-04-12

**Authors:** Megha Sharma, Rishika Bachani

**Affiliations:** 1Department of Pharmacology, R. D. Gardi Medical College, Ujjain 456006, MP, India; 2Department of Global Public Health-Health Systems and Policy: Medicines, Karolinska Institutet, 171 77 Stockholm, Sweden; 3R. D. Gardi Medical College, Ujjain 456006, MP, India

**Keywords:** standard precautions, post-exposure prophylaxis, know–do gap, future healthcare professionals, barriers to compliance with standard precautions, biomedical waste management, knowledge, attitude, and practice, hygiene, nursing students, medical students

## Abstract

Objectives: The aim of this study was to assess and compare (a) the knowledge, attitude, and practice of standard precautions (SPs), (b) the knowledge of post-exposure management, and (c) the perceived barriers underlying the noncompliance with SPs among future healthcare professionals (HCPs), i.e., students of medical and nursing courses in Central India. Setting and participants: A cross-sectional study was conducted in 2017–2018 among students of a medical and a nursing college using a pretested and modified questionnaire. Data were collected during 23 face-to-face sessions. Responses were scored according to standard guidelines of the Centers for Disease Control and Prevention and WHO, where each correct response was given a score of 1. Results: Among 600 participants, 51% of medical students and 75% of nursing students could not select the correct definition of SPs from the given options. Sixty-five percent of medical students (275/423) and 82% of nursing students (145/177) were unaware of the term post-exposure prophylaxis. Overall, knowledge about personal protective equipment and hazard symbols was poor (<25%). Furthermore, although theoretical knowledge about hand hygiene was good (510/600; 85%), its implementation was poor (<30%). Sixty-four percent of participants believed that the use of hand rub replaced the need for handwashing, even for visibly soiled hands. Some of the participants believed that the use of PPE might offend patients (16%). High workload and poor knowledge were other significant barriers underlying the noncompliance with SPs. Conclusions: A suboptimal translation of participants’ knowledge into practice is evident and signifies the presence of the know–do gap. Poor knowledge and inappropriate presumptions about the use of SPs discourage the practice of SPs. This results in increased healthcare-associated infections, increased treatment costs, and a suppressed social economy. The inclusion of a dedicated curriculum with repeated hands-on and practice-based training on SPs is suggested to minimize this know–do gap among future healthcare workers.

## 1. Introduction

### 1.1. Background

Healthcare-associated infections (HAIs) are defined as those that appear in patients within 48 h of admission to a healthcare facility [1,2]. However, it is often overlooked that not only patients but also caregivers, including healthcare professionals (HCPs), are at risk. The HAIs occurring among HCPs are commonly known as occupational infections. Some of these occupational infections can be fatal, such as hepatitis B, hepatitis C, tuberculosis, and human immunodeficiency virus (HIV) [3,4,5,6]. HAIs result in extended hospital stay, healthcare costs, morbidity, economic loss, work hours lost, and an increase in the use of antibiotics, leading to increased antibiotic resistance and mortality [7,8,9].

Low- and middle-income countries (LMICs) are disease-endemic, resource-constrained countries with high infection burdens and a high risk of HAIs [10,11]. The lack of HAI surveillance systems is also affected by the complexity and lack of standardized criteria to diagnose infections. Moreover, LMICs do not have surveillance systems to register and monitor occupational infections. As a result, the occupational safety and prevalence of infectious diseases remain neglected and highly underreported [11,12,13,14]. Therefore, it is a challenge to obtain reliable information on HAIs from all countries. The results from the available studies indicate that hundreds of millions of patients are affected by healthcare-associated infections each year globally, and, even though occupational infections are underreported, more than 90% of the global HAIs are reported from LMICs [3,15]. According to the World Health Organization (WHO), over three million HCPs are infected each year with occupational infections while serving in healthcare facilities [15]. Occupational infections are the HAIs developed due to unprotected close contact with infected sources such as infected patients, needles, infected cotton swabs, mucus membranes, and surgical instruments or the contaminated environment of a healthcare facility [3,16,17].

HAIs can be minimized and controlled using gold-standard measures, such as standard precautions (SP). According to the Centers for Disease Control and Prevention (CDC), SPs are defined as the minimum infection prevention practices to be applied during patient care in a healthcare facility, regardless of the suspected or confirmed infection status of the patient to prevent HAIs [18,19]. The seven components of SP are (i) hand hygiene (HH), (ii) use of personal protective equipment (PPE) such as gloves, masks, and eyewear, (iii) respiratory hygiene/cough etiquette, (iv) sharps safety under proper handling of bio-medical waste (BMW), (v) safe injection practices (i.e., an aseptic technique for parenteral medications), (vi) use of sterile instruments and devices, and (vii) clean and disinfected environmental surfaces of healthcare facility [18,19,20].

Knowledge of SPs is introduced as a part of the undergraduate course curriculum in pre-final professional year (third year of the Indian medical education curriculum) of the MBBS and in the first and second years, respectively, of the graduate nonmatriculated nursing and bachelor of nursing courses [21,22]. However, the failure to gain knowledge and exercise SPs during undergraduate years might lead to the poor practice of SPs in the future, leading to increased HAI rates [21,22]. Moreover, the gap in knowledge and practice is another issue to investigate. Few studies have assessed the knowledge of and compliance with SPs among practicing HCPs [22,23,24,25]; however, none were conducted among students of healthcare courses (MBBS and nursing). A majority of these studies described the poor knowledge and practice of HH and infection prevention measures among practicing HCPs [2,13,14]. Despite the fact that students of healthcare courses (MBBS and nursing) are future HCPs, none of the studies assessed the knowledge, attitude, practice, and perceived barriers of SPs among these groups.

### 1.2. Objectives

The aim of this study was to assess and compare (a) the knowledge, attitude, and practice of standard precautions, (b) the knowledge of post-exposure management, and (c) perceived barriers underlying the noncompliance with standard precautions among future healthcare professionals, i.e., students of medical and nursing courses in Central India.

## 2. Material and Methods

### 2.1. Study Setting and Participants

This cross-sectional questionnaire-based study was conducted in 2017–2018 at R.D. Gardi Medical College among students of the Bachelor of Medicine and Bachelor of Surgery (MBBS) courses, as well as at R.D. Gardi College of Nursing, Ujjain, India [26].

A comprehensive approach was adopted to provide an equal opportunity for all students to participate in the study. All MBBS students enrolled at R.D. Gardi Medical College, as well as Bachelor of Science in Nursing (BSN) and Graduate Nonmatriculated Nursing (GNM) students enrolled at R.D. Gardi College of Nursing, were invited to participate voluntarily in the study. The invitations were sent through printed flyers, announcements, and verbal communication. The data were collected using a contextually designed questionnaire as a data collection tool.

#### 2.1.1. Inclusion Criteria

An all-inclusive approach was adopted for data collection from the students enrolled in R.D. Gardi Medical College and R. D. Gardi College of Nursing.The students who attended scheduled clinical postings as per the medical and nursing curricula and had direct contact with the patients (MBBS students in the second year onward).

#### 2.1.2. Exclusion Criteria

Students who were not willing to participate or who did not give written consent for participation.First-year MBBS students (due to non-exposure to the clinical area).

### 2.2. Data Collection

All possible efforts were made to include maximum participation. Twenty-three face-to-face sessions were held for data collection, at the convenience of the participants and researcher. The aim and objectives of the study were explained to the participants, and written informed consent was obtained from each participant before data collection.

### 2.3. Data Collection Tool

A specifically designed data collection tool was used that could be filled in 20–30 min. Most of the questions were based on standard definitions or the recommendations of global organizations such as the CDC and WHO [2,3,4,5,6,7,8,9,10,11,12,13,14,18]. The printed questionnaire included a mix of closed (yes/no), structured (multiple-choice), and semi-structured/open-ended questions. The questionnaire included a mix of structured and semi-structured items under the following segments: (a) sociodemographic details including name, age, gender, religion, course, year of study in respective curriculum, batch, information about previous training or orientation attended on infection prevention control, and management of post-exposure risk; (b) knowledge about SPs and its main components, as well as measures to be taken for infection prevention and control, and for managing post-exposure prophylaxis (PEP); (c) attitude toward SPs, PEP, and HAIs; (d) self-reported practice and barriers toward compliance with infection prevention and control, SPs, and PEP.

The questionnaire contained subsections of knowledge, attitude, and self-reported practices and barriers under the following headings: general concept of SPs and infection control, PPE, hand hygiene (WHO-based hand hygiene recommendations), disposal of BMW and management of sharp objects, handling of patient care equipment, PEP, and perceived barriers responsible for noncompliance with SPs.

All questions were printed in English and Hindi for ease of understanding for the participants. The English questions were translated into Hindi and retranslated into English, and vice versa, to affirm their reliability and accuracy. The same questionnaire was used among both participant groups to ensure the comparability of the results.

The questionnaire was specifically designed to measure knowledge, attitudes, and practices related to SPs and PEP. The questionnaire was pretested (data not included in the present study) to check its comprehensibility and validity. The final questionnaire consisted of 60 analyzable points (questions) including a few subsections with probing questions. All responses to the open-ended questions were coded to facilitate the quantitative analysis. Questionnaires with a response rate of ≥40% were included in the analysis. Most of the participants appeared in small groups. The aim and objectives of the study were explained, and written informed consent was obtained from each participant prior to data collection. Participant responses to the questionnaire were recorded in 23 face-to-face sessions and scored according to the guidelines of the CDC and the WHO.

Multiple questions on the same topic were framed on the basis of standard definitions and recommendations from the CDC and WHO to endorse their validity. The inclusion of both closed and open-ended questions allowed a comprehensive assessment of participants’ understanding and attitudes, as well as verified their reliability. Moreover, the use of face-to-face sessions provided an opportunity for clarification, increasing the validity, accuracy, and completeness of the responses.

### 2.4. Methodological Considerations

The CDC definition of SPs was considered as the correct definition, which states that SPs are infection prevention measures that essentially include hand hygiene (HH), use of personal protective equipment (PPE), and proper handling of biomedical waste (BMW) [18].

Each correct response was given a score of 1, whereas wrong or inappropriate responses were given a score of 0. A total of 35 points could be scored for correct responses to the knowledge questions, along with seven points for correct practice and 14 points for appropriate attitude. Binary variables were generated for the closed questions (yes/no), and the multiple-choice answers were entered as nominal categorical variables. All responses were evaluated according to the CDC and WHO guidelines [18,20]. The scores were divided into four levels: poor (<40%), average (41–60%), good (61–80%), and outstanding (81–100%) [18,20]. Overall scores of knowledge, practice, and attitude were calculated by combining the scores of all knowledge-, practice-, and attitude-related questions, respectively.

As part of an ongoing, long-term research project, an in-house prepared alcohol-based hand rub (ABHR) was introduced in the study hospital associated with the medical and nursing colleges [27]. This interventional project included multiple workshops to improve HH compliance in the colleges [27]. The workshops were organized primarily for the HCPs, but the students were also invited to participate in at least one such workshop. Participation in the workshops was voluntary.

### 2.5. Data Management and Analysis

Data were entered in EPI Info 3.1 and analyzed using STATA V.13.1 (Stata Corp, College Station, TX, USA). Each participant was given a unique code to maintain full confidentiality during the analysis. Frequencies and percentages were counted and calculated for categorical variables, and the chi-square test was used to assign the significance of *p*-values for the participant groups. A *p*-value < 0.05 was considered significant. Tables present the results of correct responses.

### 2.6. Ethics Statement

The study was approved by the ethics committee of R.D. Gardi Medical College, Ujjain (approval number 69/2016). Written consent was obtained from all participants after explaining the study’s objectives, procedures, and ethical considerations. The participants were assured that full confidentiality would be maintained for the responses provided during data analyses. It was explained that participants would not get any direct benefits or compensation for participating in the study, nor would there be any impact on their studies or academic grades if they decided not to participate.

## 3. Results

All students enrolled in two healthcare courses (704 students; MBBS = 491 and nursing = 213) were approached and invited to participate in the study. Of the total, 604 students showed interest to participate in the study (604/704 = 86%, MBBS = 424 and nursing = 180). Out of the interested participants (n = 604), four participants did not respond to the survey, resulting in a final sample of 600 participants (MBBS = 423 and nursing = 177). The nursing group consisted of more female participants (73%) compared to the MBBS group, which had a female proportion of 47% (*p* < 0.05). Forty-seven percent of MBBS students and 14% of the nursing students reported having previously attended training on HH.

### 3.1. Knowledge about Standard Precautions and Preventive Measures

The overall knowledge of SPs was poor among both participant groups. Fifty-one percent of the MBBS students and more than 75% of nursing students could not select the correct definition of SPs from the options provided in the questionnaire (Question 2, Table 1) [28]. On the contrary, the term SPs was defined as paying respect to the patients, providing psychological support to patients, sterilizing instruments, and daily bathing. The overall knowledge about the WHO’s recommendations on HH was also poor among both groups (<30%) but higher among MBBS students than nursing students (Question 4, Table 1, *p* < 0.05) [20]. Twenty percent of MBBS students and 1% of nursing students knew about WHO’s recommended “five” moments of HH (Question 4a, Table 1). More than 80% of the participants knew the moments of washing hands with soap and water (Question 5a, Table 1).

Sixty-five percent of MBBS and 82% of nursing students did not know when PEP should be applied (Question 6, Table 1). The effectiveness of antiretroviral therapy (ART) as PEP was recognized by 30% of the participants in both groups (Question 7, Table 1). Fewer nursing students (14%) than MBBS students (24%, Question 8, Table 1), selected “N95 respirator” as the most appropriate PPE to maintain respiratory hygiene during aerosol-generating procedures against airborne diseases. However, gloves and aprons were selected by more than half of the nursing students (53%) as the most appropriate PPE to ensure respiratory hygiene (Question 8, Table 1).

The participants had scanty knowledge about the segregation of biomedical waste (BMW) and the management of BMW based on color-coded bins (Question 10, Table 1). Sixty-six percent of MBBS students and 78% of nursing students believed that BMW could be discarded by regular municipal disposal systems. None of the nursing students could recognize the hazard symbols used for radioactive and explosive materials (Question 11, Table 1). Moreover, the symbol for “radioactive material” was identified as an exhaust fan or wind energy by 18% of the MBBS and 30% of nursing students, and the symbol of “nonionizing radiation” was identified as a symbol for sound, Wi-Fi, or Internet hotspot by 20% MBBS and 31% nursing students. The difference in mean overall knowledge scores among male and female participants was not significant and ranged between 12.74 for male nursing participants and 13.28 for female MBBS participants.

Figure 1 presents the average knowledge percentages of various components of SPs among both study groups. The knowledge of more than 80% of participants was good about HH, whereas less than 35% of the participants had correct knowledge about handling sharps and appropriate use of PPE and PEP.

### 3.2. Attitudes toward the Use of SPs

Fifty-four percent of the MBBS students and 73% of nursing students believed that ABHR replaces the need for washing hands with soap and water even if the hands are visibly soiled (Question 4a, Table 2). For the question about the disposal of used needles, significantly fewer MBBS students (13%) compared to nursing students (62%) opted to use a needle cutter (Question 2, Table 2). For a situation of needlestick injury, 41% of MBBS and 75% of nursing students said that they would notify nursing staff about the injury. Twenty-one percent of MBBS and 34% of nursing students believed that all HCPs were at high risk for HAIs (Question 5, Table 2). Forty-four percent of MBBS students and 51% of nursing students could not list the infections potentially caused by accidental needle stick injury (Question 6, Table 2). The most common HAIs mentioned were HIV, acquired immunodeficiency syndrome (AIDS), hepatitis, and tetanus.

### 3.3. Practice of SPs

The participants were given six hypothetical cases to manage, reflecting their views on the ideal practice of SPs (Questions 10 to 15, Table 2). For a hypothetical case of a bleeding patient in the emergency department, a significantly higher number of MBBS students (70%) than nursing students (47%) opted to press the bleeding site without using any PPE (gloves). More than half of the participants understood that advice should be sought from a pulmonary consultant if a patient with a pulmonary infection coughed and spat toward the face of an HCP during the examination. Most of the participants placed themselves in a range of average to good SP practice zone (self-reported practice); however, the scores were calculated on the basis of responses to practice-related questions, whereby most of the participants appeared in the poor range of SP practice (Figure 2).

### 3.4. Perceived Barriers Underlying Noncompliance with SPs

Lack of knowledge about appropriate PPE and high workload were perceived as influential barriers underlying noncompliance with SPs (Figure 3). According to Figure 3, 19% of MBBS students and 13% of nursing students believed that the use of PPE might offend the patient. In an emergency situation, patient care was prioritized over practicing SPs by 39% of the MBBS students. Seventy-three percent of MBBS and half of the nursing students believed that all patients were not a source of infection. According to Table 2 (Questions 1 and 2), a higher percentage of MBBS students than nursing students expressed their willingness to attend a training/workshop focusing on SPs.

## 4. Discussion

Our study highlighted that the theoretical knowledge about HH was significantly higher in both groups of future HCPs; however, the transformability of this knowledge into practice was poor. Both participant groups had deficient knowledge about SPs and poor competence with regard to the infection risk assessment. Half of the MBBS students and more than three-quarters of the nursing students could not select the correct definition of SPs. Moreover, they could not identify the primary constituents of SPs (namely, HH, use of PPE, and proper disposal of BMWs). Most of the participants of both groups were unaware of the common terms and abbreviations related to the components of SPs, as well as symbols of hazards. The overall knowledge score of the nursing students was lower than that of the MBBS students. Misconceptions such as ABHR replacing the need for handwashing and beliefs that SP might offend patients were also evident in both groups. As reflected in our study, the scarcity of appropriate knowledge about SPs in future HCPs, can potentially contribute to increases in the rates of HAIs, treatment duration, and use of antibiotics. This will ultimately constrain the healthcare system, resources, and the economy [14,29].

HH is considered a gold standard to prevent the spread of infections, representing one of the most cost-effective and crucial components of SP [17,18]. The knowledge score for the use of recommended WHO HH techniques (how and when to wash/rub hands) was observed to be at an outstanding level [20,30]. A high score for participant knowledge of hand hygiene could be attributed to the training conducted as part of an interventional study [27]. However, few participants knew about the correct HH techniques, and the knowledge did not translate into practice. This supports the globally reported know–do gap [7,30,31,32]. Furthermore, most participants were unaware of the importance of washing their hands with soap and water, even if their hands were visibly soiled. The misconception that ABHR replaces the need for handwashing is a serious point of concern that can significantly contribute to the development of HAIs.

Risk assessment for HAIs is the first step in selecting an appropriate PPE [18]. Most of the participants could not assess the level of infection risk. The participants acknowledged the sputum of a tuberculosis patient and blood of an HIV patient as being a risk for HAIs while they overlooked the potential risk from unrecognized sources such as soiled PPE (MBBS, 65%; nursing, 88%). In our study, fewer participants (39%) believed that every patient should be considered as a source of infection irrespective of the clinical condition of the patient. Our study found a lower percentage (39%) compared to a study among medical students in Saudi Arabia (42% of students), who believed every patient should be considered a potential source of infection, irrespective of their clinical condition [33]. More than half of the participants could not identify the most common types of blood-borne infections that might occur due to injuries caused by sharps. This knowledge was lower among our participants compared to studies conducted in West Bengal and Southwest Ethiopia [34,35]. It is to be noted that the Ethiopian study was conducted among HCPs and not among future HCPs. Higher knowledge and better practice related to infection prevention and control are expected among HCPs than among students of healthcare courses. However, the poor knowledge and inadequate risk-assessment aptitude among our participants, compared to available studies, raise concerns.

Universally accepted hazard symbols are used as a caution for possible exposure hazards and a reminder to adopt precautionary measures while handling hazardous substances. None of the nursing students and too few MBBS students could recognize the symbols for radioactive material and explosives. Moreover, the identification of hazard symbols displayed in the questionnaire was incorrect. For example, the symbol of radioactive material was identified as an exhaust fan and wind energy, while the symbol of nonionizing radiation was identified as Wi-Fi or Internet hotspot. Poor or inaccurate knowledge about these hazard symbols can put the entire healthcare facility and nearby population in life-threatening situations. Immediate interventions are needed to impart awareness to eliminate misconceptions and to identify the infection risk to ensure safe occupational practices in future HCPs. Strategic and repeated training for HCPs and undergraduate students could be beneficial to achieve this goal.

The selection of appropriate PPE is essential to protect oneself from potentially infectious materials. A majority of the participants could not classify respiratory infections as the most prominent exposure risk for a hypothetical case of an airborne disease. Most of the participants (69%) selected gloves or an apron as the most appropriate PPE, rather than the “N95 respirator” for a case of an air-borne disease [13]. For the given presumptive situations, treating the patient was prioritized over the use of PPE. Proper selection and use of appropriate PPE during patient care, even in emergencies, are essential, and failure to comply may result in post-exposure consequences [36].

Noncompliance with PPE among HCPs was observed in our study, highlighting that perceived barriers and misleading beliefs contribute to noncompliance with SPs. In our study, the majority of the participants indicated workload as the main barrier underlying noncompliance with SPs, followed by lack of knowledge about PPEs, poor infection risk assessment, and inconvenience of using PPE. Additionally, we found that 16% of participants believed that the use of PPE might offend the patients and, therefore, that PPE should not be used. This misconception could lead to feelings of guilt and hinder compliance with safe clinical practices, potentially harming the community in future. Therefore, it is crucial to address and correct this misconception.

BMW is defined as highly infectious waste generated at a healthcare facility that requires proper segregation and disposal [37]. The participants had poor knowledge about BMW segregation and proper sharps disposal, particularly among MBBS students. The next step after the segregation of BMW is proper disposal. However, most participants were unaware of the protocol for appropriate BMW disposal, posing yet another potential threat to the handler, coworkers, the environment, and the community [38,39]. The nursing course curriculum includes tasks such as bedmaking, dressing, and minor clinical procedures, allowing nursing students to actively participate in BMW management. In contrast, the current MBBS curriculum does not include independent clinical practice during the education period; thus, medical students have limited access to BMW disposal areas. This may explain why nursing students had a higher BMW-related knowledge than medical students [31]. Thus, the poor knowledge of BMW disposal might be due to the minimal participation of medical students in BMW management [31].

According to National AIDS Control Organization (NACO) guidelines, all cases of accidental exposure to blood or other contaminated fluid of a patient through a cut or needle stick should be notified to the ART center at the earliest convenience (within 72 h), regardless of whether the patient is a PLWHA (person living with HIV/AIDS) or their status is unknown [40]. Ironically, most of the participants were unaware of the effective period to start PEP and stated that they would notify the nursing staff, instead of reporting to the ART center. Unawareness about whom to notify after exposure and the crucial time to start PEP medication were common features in both participant groups but were more prevalent among the nursing students [41]. The lack of knowledge about PEP can escalate the severity of the infection and decrease the benefits of antiretroviral medications [42,43]. Thus, it is important to impart knowledge about PEP to future HCPs.

The results of our study indicate that there is insufficient focus on explaining the benefits of prevention mechanisms and the consequences of HAIs in the respective curricula of medical education. This issue is under-focused globally and needs immediate attention. The student phase of life is a crucial learning period, and recognizing knowledge gaps is essential for bringing about desired changes. Participants of our study might have recognized their knowledge gap and demonstrated their interest in attending workshops related to SPs in future. Therefore, more emphasis must be placed on practicing the components of SPs during undergraduate and postgraduate education periods.

## 5. Strengths and Limitations

The present study used a cost-effective, feasible, adaptive, and appropriate design to achieve its aims. Data were collected by a peer (student); this might have minimized the pressure of social desirability on the participants and helped to register unbiased responses. The same questionnaire was used to improve the comparability. Although the flexibility to participate in small groups in face-to-face sessions prolonged the duration of data collection, this convenience might have contributed to better reliability and a high response rate (>85%). The questionnaire was printed in English and Hindi. This might have provided a better opportunity to understand and respond to the questions.

Our study had a few limitations. Responses to the hypothetical/futuristic practice questions were reported by the students, but the actual practice was not observed. However, probing for self-reported practice is an inexpensive method, and this was considered the best approach to achieve the aims of the study. The inclusion of first-year students (with no clinical exposure) might have provided a complete view of MBBS students; however, since the topic of SP is not introduced in the first-year curriculum, they were excluded. There is a possibility that a reflection of the workshops conducted during the interventional project might have been seen in the results of hand hygiene-related questions in the present study. Lastly, the instrument used to assess the KAP was not statistically validated and tested for reliability. However, multiple questions on the same topic were framed on the basis of standard definitions and recommendations from the CDC and WHO to endorse their validity. The inclusion of both closed- and open-ended questions allowed a comprehensive assessment of participants’ understanding and verified their reliability. Moreover, the use of face-to-face sessions provides an opportunity for clarification to increase the validity, accuracy, and completeness of the responses.

## 6. Conclusions and Future Implications

Our results show that, although the participants had good theoretical knowledge about HH, the transformability of this knowledge into practice was poor. The absolute lack of knowledge of the hazard symbols was unexpected and could place the community at large risk. This study found that both participant groups had misconceptions about infection control practices, such as the belief that ABHR replaces handwashing and that using PPE might offend patients. Moreover, the overall knowledge about the components related to SP and competence in infection risk assessment was poor among both participant groups. Barriers such as workload, unavailability of PPE, lack of knowledge, and misconceptions were considered as the main factors responsible for noncompliance with SPs. The scarcity of appropriate knowledge about infection prevention and control among future HCPs can potentially contribute to increased rates of HAIs and the use of antibiotics, which can constrain the healthcare systems, resources, and economy.

Immediate interventions are needed to impart awareness to eliminate misconceptions and identify the infection risk to ensure safe occupational practices for future HCPs. Strategic and repeated training for future HCPs could be beneficial to achieve this goal. The misconceptions should be addressed and corrected during training. Raising awareness of these potential consequences could encourage healthcare professionals to prioritize infection control practices.

Recognizing knowledge gaps and understanding the benefits of prevention mechanisms, as well as the potential consequences of HAIs, represent the first steps toward improving SP practices. Therefore, it is important to include infection risk assessment as a key component of infection control training. Providing workshops and training sessions accompanied by regular observation and feedback can help bridge the knowledge gap and improve SP practices among the HCPs. A majority of our participants showed a high willingness to attend SP workshops and HH training in future. This receptiveness shows participants’ concern to improve their knowledge about SPs. Thus, more focus should be given to SP components during medical education globally, especially during the undergraduate period.

Lastly, it is important that medical and nursing schools globally address the knowledge gaps identified in this study to ensure that future HCPs are adequately prepared to prevent and control HAIs. This will motivate both present and future HCPs to improve their practice of and compliance with SPs. This will ultimately lead to improved patient outcomes, lower treatment burden, and safer working environments for HCPs.

## Figures and Tables

**Figure 1 ijerph-20-05487-f001:**
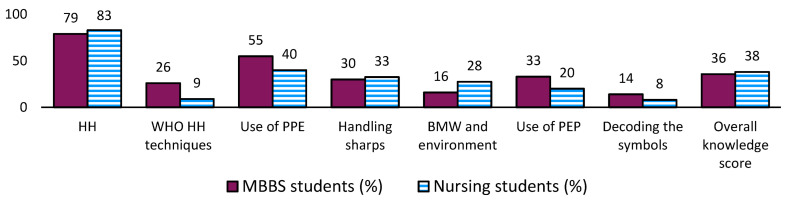
Knowledge about standard precautions among the participants pursuing nursing and MBBS courses in Ujjain, Central India. Knowledge score classification: poor, <40%; average, 41–60%; good, 61–80%; outstanding, 81–100%. Abbreviations: BMW, biomedical waste; HH, hand hygiene; MBBS, Bachelor of Medicine and Bachelor of Surgery; PEP, post-exposure prophylaxis; PPE: personal protective equipment; WHO, World Health Organization.

**Figure 2 ijerph-20-05487-f002:**
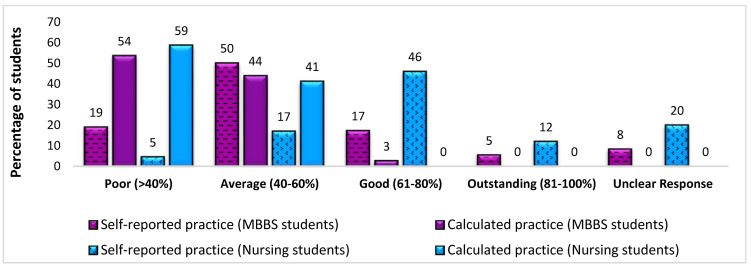
Comparison of percentages of participants’ self-reported and calculated scores for the practice of standard precautions. MBBS—Bachelor of Medicine and Bachelor of Surgery.

**Figure 3 ijerph-20-05487-f003:**
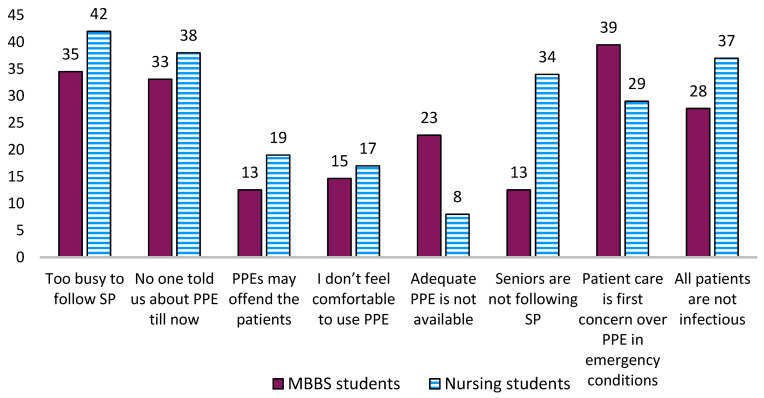
Perceived barriers underlying noncompliance with standard precautions and use of PPE by the participants pursuing nursing and MBBS courses in Ujjain, Central India. MBBS—Bachelor of Medicine and Bachelor of Surgery, PPE—personal protective equipment, SPs—standard precautions.

**Table 1 ijerph-20-05487-t001:** Knowledge of the future HCPs on standard precautions.

	Questions with Correct Answers	Total N = 600; n (%)	MBBS N = 423; n (%)	Nursing N = 177; n (%)	*p*-Value
1.	Select the correct definition of standard precautions (select correct definition) ^1^	250 (48)	209 (49)	41 (23)	<0.05
2.	Select the correct statements				
	(a) Washed gloves cannot be reused. (Correct)	336 (56)	258 (61)	78 (44)	<0.05
	(b) Cotton or gauze masks are not advantageous over surgical masks. (Correct)	290 (48)	225 (53)	65 (37)	<0.05
	(c) All types of body fluids of a patient are considered infectious. (Correct)	269 (45)	166 (39)	103 (58)	<0.05
	(d) All patients must be considered as potential sources of infection. (Correct)	203 (34)	115 (27)	88 (50)	<0.05
3.	Select the correct option based on the WHO hand-hygiene recommendations	126 (21)	111 (26)	15 (9)	<0.05
	(a) There are five moments of HH based on “my 5 moments for hand hygiene”.	86 (14)	84 (20)	2 (1)	<0.05
	(b) There are six steps to complete HH with ABHR.	171 (29)	163 (39)	8 (5)	<0.05
	(c) There are eight steps in the standard handwashing technique.	70 (12)	52 (12)	18 (10)	>0.05
	(d) The time required for washing hands with soap and water is 40–60 s.	166 (28)	142 (34)	24 (14)	<0.05
	(e) The time required for hand rub using an ABHR is 20–30 s.	139 (23)	115 (27)	24 (14)	<0.05
4.	Correct knowledge of when to perform hand hygiene	481 (80)	334 (79)	147 (83)	
	(a) It is necessary to wash hands with soap and water	478 (80)	327 (77)	151 (85)	<0.05
	(i) Before wearing gloves.	424 (71)	275 (65)	149 (84)	>0.05
	(ii) After removing gloves.	555 (93)	390 (92)	165 (93)	>0.05
	(iii) After contact with inanimate objects near a patient.	553 (92)	387 (91)	166 (94)	>0.05
	(iv) After touching body fluids, or contaminated items, even if gloves were worn	513 (86)	365 (86)	148 (84)	>0.05
	(v) Before handling invasive devices.	486 (81)	342 (81)	144 (81)	>0.05
	(vi) After a general examination of an anemic patient.	343 (57)	208 (49)	135 (76)	<0.05
	(b) Gloves should be changed while dressing two sites on the same patient.	431 (72)	321 (76)	110 (62)	>0.05
	(c) Hand ornaments should be removed at the workplace.	550 (92)	387 (91)	163 (92)	<0.05
5.	The full form of PEP is “post-exposure prophylaxis”.	174 (29)	155 (37)	19 (11)	<0.05
6.	PEP is performed after exposure to infectious material but before onset of the disease.	178 (30)	147 (35)	31 (18)	<0.05
7.	Antiretroviral drugs are the only effective medicines for PEP.	181 (30)	128 (30)	53 (30)	>0.05
8.	An N95 respirator is the best PPE to protect against infection by airborne diseases.	127 (21)	102 (24)	25 (14)	<0.05
9.	Biomedical waste is segregated at the study hospital.	290 (48)	170 (40)	120 (68)	<0.05
10.	The correct color of the bins used to dispose of biomedical wastes				
	(a) Infectious wastes: bandages, gauze, and cotton swabs are disposed of in the yellow bin.	123 (21)	63 (15)	60 (34)	<0.05
	(b) The patient’s body fluids and wastes are disposed of in the yellow bin.	103 (17)	38 (9)	65 (37)	<0.05
	(c) All types of glass (bottles and broken glass articles) are disposed of in the blue bin.	102 (17)	62 (15)	40 (23)	>0.05
	(d) Needles, syringes, razor blades, and metal articles are disposed of in the black bin.	43 (7)	30 (6)	13 (7)	>0.05
	(e) Plastic catheters, syringes, and intravenous fluid packages are disposed of in the red bin.	82 (14)	45 (11)	37 (21)	<0.05
	(f) Expired medicines are disposed of in the blue bin.	16 (3)	9 (2)	7 (4)	>0.05
	(g) Used PPE cannot be discarded by regular municipal disposal systems.	183 (31)	144 (34)	39 (22)	<0.05
11.	Decode the hazard symbols				
	(a) 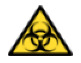 (Biohazard)	76 (13)	74 (17)	2 (1)	<0.05
	(b) 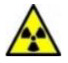 (Radioactive material)	60 (10)	60 (14)	0 (0)	<0.05
	(c) 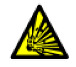 (Explosive)	56 (9)	56 (13)	0 (0)	<0.05
	(d) 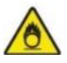 (Inflammable)	231 (39)	179 (42)	52 (29)	<0.05
	(e) 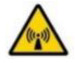 (Nonionizing radiation)	95 (16)	93 (22)	2 (1)	<0.05

N—total number of participants; n (%)—the number of participants (percentage). Abbreviations: ABHR—alcohol-based hand rub; HCPs—healthcare professionals; HH—hand hygiene; MBBS—bachelor of medicine and bachelor of surgery; WHO—World Health Organization; PEP—post-exposure prophylaxis, PPE—personal protective equipment. ^1^ As per the CDC [18].

**Table 2 ijerph-20-05487-t002:** Attitudes and self-reported practices of the future HCPs regarding the use of the standard precautions.

	Expected Correct Responses to the Questions	Total n (%)	MBBS n (%)	Nursing n (%)	*p*-Value
	Attitude and Beliefs	N = 600	N = 423	N = 177	
1.	(a) Previously attended HH training and would also like to attend training in future.	188 (31)	169 (40)	19 (11)	<0.05
	(b) Did not previously attend any training but would like to attend in future.	265 (44)	169 (40)	96 (54)	<0.05
2.	A used needle should be disposed of by separating the needle with a needle cutter.	165 (28)	55 (13)	110 (62)	<0.05
3.	An episode of needle stick injury should be notified to the ART center.	116 (19)	91 (22)	25 (14)	>0.05
4.	(a) ABHR does not replace the need for handwashing in any condition.	243 (41)	195 (46)	48 (27)	<0.05
	(b) Use of gloves does not replace the need for HH.	349 (58)	320 (76)	102 (58)	>0.05
	(c) Used needles should not be recapped before disposal.	349 (58)	320 (76)	29 (16)	<0.05
	(d) Used needles must be bent with a needle cutter after use.	488 (81)	343 (81)	145 (82)	>0.05
	(e) Soiled sharp objects must be shredded before final disposal.	453 (76)	325 (77)	128 (72)	>0.05
	(g) Family members/nursing staff can remind the doctor to perform hand hygiene.	451 (75)	312 (74)	139 (79)	>0.05
5.	HCPs are at high risk while treating patients and during their clinical postings	147 (25)	87 (21)	60 (34)	<0.05
6.	List the infections that can be caused due to needle stick injury (HIV/AIDS, hepatitis B, and hepatitis C).	365 (61)	279 (66)	86 (49)	<0.05
7.	The present study has sensitized me to the importance of SPs.	351 (59)	281 (66)	70 (40)	<0.05
8.	I am not convinced that SPs are necessary.	111 (19)	45 (11)	66 (37)	<0.05
	Practice (correct practice)	Total n (%)	MBBS n (%)	Nursing n (%)	*p*-value
		N = 600	N = 423	N = 177	
9.	List the instruments you use as standard precautions during clinical practice				
	(a) Apron	38 (6)	3 (2)	35 (8)	<0.05
	(b) Mask	71 (12)	22 (12)	49 (12)	<0.05
	(c) Handwashing/ABHR	153 (26)	32 (18)	121 (29)	<0.05
	(d) Sterilization of instruments	35 (6)	3 (2)	32 (8)	<0.05
	(e) Proper management of BMW	27 (5)	0 (0)	27 (27)	<0.05
	(f) Gloves	171 (29)	23 (13)	148 (35)	<0.05
10.	If during a lumbar puncture: accidental spilt of the cerebrospinal fluid should be promptly wiped off and the surface should be cleaned using a disinfectant, followed by proper handwashing with soap and water.	430 (72)	300 (71)	130 (73)	>0.05
11.	If a patient with pulmonary infection starts coughing and spits toward you during the examination: advice should be sought from a pulmonary consultant.	357 (60)	237 (56)	120 (68)	>0.05
12.	If a patient is about to vomit: the HCP should wear gloves and a mask and bring the kidney tray to help the patient.	476 (79)	329 (78)	147 (83)	>0.05
13.	To manage a patient with severe bleeding in an emergency department, immediate attempts should be made to prevent bleeding from the injured site by having gloves on.	221 (37)	127 (30)	94 (53)	<0.05
14.	To collect blood samples: HCPs should wash and dry hands, wear gloves and a face mask, sterilize the site, and withdraw the blood.	361 (60)	204 (48)	157 (89)	<0.05
15.	In a code blue emergency at a cardiac intensive care unit: CPR should be performed only after using the available PPE.	184 (31)	101 (24)	83 (47)	<0.05

N—total number of participants; n (%)—number of participants for question (percentage). Significant *p*-values < 0.05 between MBBS and nursing students. Abbreviations: ABHR—alcohol-based hand rub; AIDS—acquired immunodeficiency syndrome; ART—antiretroviral therapy; BMW—biomedical waste; CPR—cardiopulmonary resuscitation; HAI—healthcare-associated infections; HCPs—healthcare professionals; HH—Hand hygiene; HIV—human immunodeficiency virus; MBBS: Bachelor of Medicine and Bachelor of Surgery; PPE—personal protective equipment; SP—standard precautions.

## Data Availability

The institutional ethics committee procedure assures maintenance of full confidentiality during analyses and beyond. Therefore, a request can be made to The Chairman, Ethics Committee, R.D. Gardi Medical College, Agar Road, Ujjain, Madhya Pradesh, India 456006 (email: iecrdgmc@yahoo.in, uctharc@sancharnet.in), giving all details of the article for data availability. The IEC approval number (69/2016) must be quoted with the request.

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
