# Peer review of "Knowledge, Attitude, Practice, and Perceived Barriers for the Compliance of Standard Precautions among Medical and Nursing Students in Central India"

_ijerph, 2023, doi:10.3390/ijerph20085487_

Round 1

Reviewer 1 Report

Introduction 

There is a need to add studies examining knowledge, attitude, practice, and perceived barriers to complying with two standard precautions among healthcare care providers, including medical students. '

Method 

validity and reliability of the tools need to explain further 

Author Response

Response to Reviewer 1 Comments

Point 1: There is a need to add studies examining knowledge, attitude, practice, and perceived barriers to complying with two standard precautions among healthcare care providers, including medical students. '

 Response: Thank you for this remark. We have now updated the reference list that includes relevant recent studies, as per your comment.

Point 2: Method 

validity and reliability of the tools need to explain further 

Response: Thank you for this comment. We have now added the text as per your valuable suggestion.

Reviewer 2 Report

Introduction line 49, please rephase the statement. line 50 please first show what LMICs is in line 47. line 75 Graduate Non-Matriculated nursing make sure you use the same font.

objective: line 87 to 89 write objective in full without using abbreviations you already have too many of them.

study setting: line 97 d Graduate Non-Matriculated use the same font.

data collection tool: line 118, specify that pre-tested data collection tool. refer readers to the attached questionnaire on supplementary materials (put it under acknowledgement).

Results: well presented according to my view.

since demographic information was collected it would be good to show knowledge gap between male and female.

Conclusion of the study.

Please improve the conclusion and focus on highlighted gaps in different groups in order to structure recommendations to different colleges.

Please include recommendation based on the findings of the study.

References: ensure that your reference included is not more than 5 years old, include recent references.

The study is interesting and catches the readers attention.

Author Response

Response to Reviewer 2 Comments

Point 1: Introduction line 49, please rephrase the statement.

Response: Thank you for this comment. We have now rephrased the statement as suggested.

Point 2: line 50 please first show what LMICs is in line 47.

Response: Thank you for this point. We have now mentioned the abbreviation in previous sentence.

Point 3: line 75 Graduate Non-Matriculated nursing make sure you use the same font.

Point 4: study setting: line 97 d Graduate Non-Matriculated use the same font.

Response: Thank you for this comment. We have now corrected the font sizes at both places as suggested.

Point 5: line 87 to 89 write objective in full without using abbreviations you already have too many of them.

Response: We agree with the reviewer and have removed all abbreviations from the objective of the study.

Point 6: data collection tool: line 118, specify that pre-tested data collection tool. refer readers to the attached questionnaire on supplementary materials (put it under acknowledgement).

Response: Thank you for this comment. We have now added the text as suggested.

Point 7: Results: well presented according to my view.

Response: Thank you for this encouraging comment.

Point 8: since demographic information was collected it would be good to show knowledge gap between male and female.

Response: Thank you for this comment. We agree that gender-wise comparison of knowledge is of interest. We have added the result in the manuscript as per suggestion. However, a majority of the participants in the nursing course were females (73%), and therefore, the difference in gender-wise results was not significant in the present data set.

Point 9: Conclusion of the study.

Please improve the conclusion and focus on highlighted gaps in different groups in order to structure recommendations to different colleges.

Please include recommendation based on the findings of the study.

Response: Thank you for the guidance. We have thoroughly reformulated the conclusion and recommendation section as suggested.

Point 10: References: ensure that your reference included is not more than 5 years old, include recent references.

Response: Thank you for this comment. We have extensively updated the reference list with recent studies however, a few guidelines and definitions were established some time back and are considered as Standard Guidelines, such as CDC and WHO guidelines and therefore, need to be in the reference list. Also, too few studies are conducted in the study area, i.e., India, therefore, those studies also need a position for comparability of our results in similar settings.

Point 11: The study is interesting and catches the readers attention.

Response: Thank you for this encouraging comment.